# The Health Literacy in Pregnancy (HeLP) Program Study Protocol: Development of an Antenatal Care Intervention Using the Ophelia Process

**DOI:** 10.3390/ijerph19084449

**Published:** 2022-04-07

**Authors:** Maiken Meldgaard, Rikke Damkjær Maimburg, Maiken Fabricius Damm, Anna Aaby, Anna Peeters, Helle Terkildsen Maindal

**Affiliations:** 1Department of Public Health, Aarhus University, 8000 Aarhus, Denmark; aaby@ph.au.dk (A.A.); htm@ph.au.dk (H.T.M.); 2Department of Obstetrics and Gynecology, Aarhus University Hospital, 8000 Aarhus, Denmark; rmai@clin.au.dk (R.D.M.); maikda@rm.dk (M.F.D.); 3Department of Clinical Medicine, Aarhus University, 8000 Aarhus, Denmark; 4School of Nursing and Midwifery, Western Sydney University, Locked Bag 1797, Penrith, NSW 2751, Australia; 5Institute for Health Transformation, Deakin University, Geelong, VIC 3220, Australia; anna.peeters@deakin.edu.au

**Keywords:** health literacy, inequality, intervention development, health literacy responsiveness, organizational health literacy, co-design, pregnancy, health promotion

## Abstract

A pregnant woman needs adequate knowledge, motivation, and skills to access, understand, appraise, and apply health information to make decisions related to the health of herself and her unborn baby. These skills are defined as health literacy: an important factor in relation to the woman’s ability to engage and navigate antenatal care services. Evidence shows variation in levels of health literacy among pregnant women, but more knowledge is needed about how to respond to different health literacy profiles in antenatal care. This paper describes the development protocol for the HeLP program, which aims to investigate pregnant women’s health literacy and co-create health literacy interventions through a broad collaboration between pregnant women, partners, healthcare providers, professionals, and other stakeholders using the Ophelia (Optimising Health Literacy and Access) process. The HeLP program will be provided at two hospitals, which provide maternity care including antenatal care: a tertiary referral hospital (Aarhus University Hospital) and a secondary hospital (the Regional Hospital in Viborg). The Ophelia process includes three process phases with separate objectives, steps, and activities leading to the identification of local strengths, needs and issues, co-design of interventions, and implementation, evaluation, and ongoing improvement. No health literacy intervention using the Ophelia process has yet been developed for antenatal care.

## 1. Introduction

Globally, social inequality is documented in the use and outcomes of antenatal care [1,2], even in social welfare states such as Denmark [3,4]. Socio-economic factors including low educational level, low income, and ethnicity are associated with a higher risk of obstetric complications and poor health outcomes for the mother and her child, including gestational diabetes mellitus, maternal stress and depression, low birth weight, preterm birth, stillbirth, and congenital malformations [2,3,4,5,6,7,8].

Health literacy is also associated with socio-economic factors [9,10] and seems to follow a social gradient; a phenomenon whereby people who are less advantaged in terms of socioeconomic position have poorer health compared to those who are more advantaged [11,12]. Several studies have investigated health literacy among pregnant women and show that health literacy levels in this group depend on e.g., employment status, ethnicity, and education [13,14]. Low health literacy might be associated with less antenatal care attendance and engagement [15,16,17], less knowledge about medication use in pregnancy [18,19], lower self-efficacy [17,20], depression [21], and smoking [22,23] among pregnant women. Existing evidence is inconsistent, and further knowledge about associations between health literacy, socio-economic factors, and pregnancy outcomes is needed.

Health literacy is a multi-dimensional concept, which can be defined as a person’s knowledge, motivation, and skills to access, understand, appraise, and apply health information to make decisions in everyday life concerning one’s own health [13,24]. Closely related to health literacy is organizational health literacy, which can be defined as “the way in which services, organizations, and systems make health information and resources available and accessible according to health literacy strengths and limitations” [25]. The World Health Organization (WHO) recommends a health literacy focus and responsiveness in preventive services to respond to inequality [26,27,28].

The Ophelia (Optimising Health Literacy and Access) process is used to guide co-design—a method to involve and engage relevant stakeholders in the process—of interventions to improve health literacy and equity in healthcare services [29]. The Ophelia process was developed and tested by Professor Richard Osborne and team in nine different primary health care settings [30]. They found it suitable as a framework to guide the generation of intervention ideas and respond to inequity in health care services. The Ophelia process has further been used as a methodological foundation for quite a few intervention projects [31,32,33], and findings show that the co-creative nature of the process can improve understanding of the needs and vulnerabilities of specific population groups in relation to health literacy [32]. A recent publication describes seven flagship European National Health Literacy Demonstration Projects (NHLDPs) conducted in different healthcare settings that focus on different non-communicable diseases but are similar in their use of the Ophelia methodological process [33].

Health literacy interventions in antenatal care could potentially improve knowledge, behavior, and ultimately reproductive outcomes. Only a few health literacy interventions have been developed specifically for pregnant women in antenatal care [34], and to the best of our knowledge, the Ophelia process has not previously been tested in antenatal care. Further research is warranted to investigate the role of pregnant woman’s health literacy-specific needs in relation to the development of effective initiatives.

The HeLP program aims to investigate pregnant women’s health literacy and co-create health literacy interventions based on local knowledge. Interventions will be developed through a broad collaboration between pregnant women, partners, healthcare providers, professionals, and other stakeholders using the Ophelia (Optimising Health Literacy and Access) process. This paper describes the development protocol for the HeLP program.

## 2. Materials and Methods

### 2.1. Setting and Study Population

The HeLP program will be conducted at two primary intervention sites, the tertiary referral hospital, Aarhus University Hospital, and the secondary hospital, Regional Hospital in Viborg. The annual number of births is approximately 5000 and 2200 at each site, respectively. The two sites handle the midwifery consultations during antenatal care. The two intervention sites differ in size, location, and organization. The inclusion of both sites is an attempt to increase the representativeness of the study population. During the development of interventions, external collaborators and other sites will potentially be identified, e.g., general practice. The study population in the HeLP program includes pregnant women referred to antenatal care at Aarhus University Hospital, Denmark, or the Regional Hospital Viborg, Denmark.

All pregnant women in Denmark have free-of-charge access to maternity care including antenatal, intrapartum, and post-partum care. The basic antenatal program is described in Figure 1 and includes three consultations in general practice and five consultations with a midwife (study sites). The first antenatal visit is normally in general practice where the pregnancy is confirmed. After the first consultation, the general practitioner forwards information after consent to the midwifery clinic affiliated with the hospital obstetric department through the woman’s electronic patient journal (EPJ). Medical, obstetric, or psychosocial factors detected in relation to the woman or the home, which entails specific attention, are normally specified in the women’s record. A specific care level classification of the pregnant woman is based on a risk assessment concerned with the woman’s history, socio-economic determinants including life circumstances, age, and obstetric, social, and mental risk factors, and is a professional judgment that may be changed during the pregnancy. The woman’s health literacy level is not systematically evaluated.

The Danish Health Authority recommends a four-level division of maternity care to secure the necessary support in relation to obstetric, social, and mental risk challenges [35]. The recommended four-level division is elaborated in Figure 1.

### 2.2. The Ophelia Process

The methodological foundation for the HeLP program is the Ophelia process, which was inspired by methodologies such as intervention mapping, quality improvement collaborative, and realist synthesis [36,37,38,39].

The Ophelia process provides a practical and systematic method to identify local strengths, barriers, needs and issues, and co-design of an intervention based on this local knowledge. The process is divided into three phases with separate steps and associated activities:Phase one: Identification of local strengths, needs, and issues includesStep 1. Project set-upStep 2. Data collection and/or extractionStep 3. Consultation to identify new ideasPhase two: co-design of interventionsStep 4. Intervention designStep 5. Intervention planningStep 6. Intervention development and refinementPhase three: implementation, evaluation, and ongoing improvementStep 7. Implementation and evaluationStep 8. Development of an ongoing improvement strategy

The Ophelia process in the HeLP program, including planned activities for the different phases and steps, is elaborated in Figure 2.

### 2.3. Phase One: Identification of Local Strengths, Needs, and Issues

Step 1. Project set-up

**Step** **1.**
*Focuses on identifying project focus, scope and aim of the HeLP program. The HeLP collaboration, organization, and according roles and responsibilities will be established.*


The HeLP program will be a broad collaboration between a large Danish research institution, Aarhus University, and two Danish hospitals, Aarhus University Hospital, and the Regional Hospital in Viborg. The program will be organized with a steering committee, a management team, and a working team. Representatives from each collaborating organ will be represented at each level. This study protocol and the above description of setting, study population, focus, scope, and aim were established and developed in close collaboration between the HeLP steering committee, the HeLP management team, and the HeLP working team.

Step 2. Data collection and extraction

**Step** **2.**
*Focuses on establishment of a data collection plan, data collection and extraction of data. Required ethical approvals will be obtained. Moreover, materials for the consultation workshops will be prepared.*


To identify local needs, a mapping of health literacy strengths and challenges through a comprehensive health survey, the HeLP-Questionnaire (HeLP-Q), will be carried out among the study population. The HeLP-Q will be developed specifically for the HeLP program and will include general health questions, items about socio-economic factors, and questions from already validated surveys (Cambridge Worry Scale (CWS) [40], the Edinburgh Postnatal Depression Score (EPDS) [41], Three-Item Loneliness Scale (TILS) [42], and the Health Care Climate Questionnaire (HCCQ) [43]). To measure health literacy, we will include the validated Health Literacy Questionnaire (HLQ) of 44 questions in total [44,45]. The HLQ consists of nine scales that will be analyzed separately: (1) feeling understood and supported by healthcare providers, (2) having sufficient information to manage my health, (3) actively managing my health, (4) social support for health, (5) appraisal of health information, (6) ability to actively engage with healthcare providers, (7) navigating the healthcare system, (8) ability to find good health information, and (9) understand health information well enough to know what to do. To further map digital health literacy competencies, we will include two domains from the eHealth Literacy Questionnaire (eHLQ) [46,47]. It will take approximately 20 min to fill out HeLP-Q.

To enhance participation, midwives in Aarhus and Viborg will be informed about the HeLP program and instructed to hand out participation cards to pregnant women containing an access link and QR code to either a Danish or an English edition of the HeLP-Q. In addition, posters with information about the program and access links and QR codes for the questionnaire will be placed at obvious locations in the midwifery consultation waiting area in both Aarhus and Viborg. Moreover, the HeLP program will be promoted on social media sites. One-pagers with instructions will be placed in the midwives’ staff room and researchers from the HeLP management team (M.M. and M.F.D.) will participate every second month in morning meetings at both study sites and inform midwives about the HeLP. We acknowledge that filling out HeLP-Q requires some level of health literacy. This may lead to differential participation according to health literacy levels and thereby a risk of selection bias. To accommodate this challenge, we plan to have research assistants present two days a week throughout the data collection period at the two study sites to assist pregnant women with expected lower health literacy to participate in the study. The assistance includes help with issues related to accessing, reading, understanding, and filling out HeLP-Q.

Data collected based on HLQ in the HeLP-Q will be analyzed to provide us with insights into the health literacy strengths and challenges of the participants. Cluster analysis (a statistical technique to group similar observations into clusters based on the observed values of several variables for each individual) will be used to identify sub-groups of pregnant women with similar patterns of HLQ scores. Then, case profiles will be developed based on the identified sub-groups. Participants are invited to consider intervention ideas, and how to respond to the pregnant women’s needs, based on case profiles and identified health literacy strengths and challenges.

Step 3. Response ideas consultation workshops

**Step** **3.**
*Focuses on establishing a consultation plan as well as arrangements for the consultation workshops and carrying out the consultation workshops.*


We plan to carry out two consultation workshops with pregnant women, their partners, health professionals, and other stakeholders in antenatal care (e.g., obstetricians, general practitioners, midwives, nurses, social workers, psychologists, NGO employees). The case profiles will be presented at the first consultation workshop together with additional results from HeLP-Q. These data provide the foundation for the dialogue, discussion, and idea mapping initiating the development of health literacy interventions.

As workshops are scheduled to be held in April and June 2022, and the data collection period proceeds until August 2022, the findings we present at the workshops will be based on parts of the dataset. We expect that approximately 700 women will have filled out HeLP-Q prior to the preparation of the workshop. Participants will be asked to provide informed, written consent before attending the workshops. The consultation plan and process of the workshop are elaborated in Table 1.

### 2.4. Phase Two

Similar to phase two and phase three is that phase one must be completed before we are able to fully plan the content in detail for these phases. Phases two and three will be planned based on results from HeLP-Q and the consultation workshops. In alignment with the Ophelia process, we are, however, able to describe the overall activities.

Step 4. Intervention design

**Step** **4.**
*Focuses on specifying the objectives for interventions. A rapid literature review will be conducted, and we will search for existing health literacy interventions in antenatal care. Afterwards, intervention ideas from the workshop will be matched with the HeLP program intervention objectives. An intervention or a package will be selected, and a logic model will be prepared.*


Based on the results from the workshops, the HeLP management team will confirm or adjust the focus, scope, and overall aim of the HeLP program. In alignment with the Ophelia process, a rapid literature review will be conducted to identify existing interventions in this area. This will be followed by a process where the produced intervention ideas from the consultation workshops and the newly established HeLP intervention objectives are matched. Consultation workshop participants will be invited for a second consultation workshop, where the aim is to discuss and prioritize an intervention package based on the matched intervention ideas and objectives. The HeLP management team will then develop one or more first draft logic models based on the suggested intervention package using shells similar to Figure 3. Logic models will be developed based on the theory by Taylor-Powell and Henert [49], and contain inputs, outputs, and outcomes. Logic model drafts will be presented to the HeLP steering committee, and members are invited to provide feedback. Afterwards, representatives from the two participating sites will be invited for a meeting where the logic models will be discussed, and agreements will be made. The HeLP management team will refine the final logic model based on agreements from this meeting.

Step 5. Intervention planning

**Step** **5.**
*Focuses on intervention planning. Project members, timeline, and budget will be revised and confirmed. Project milestones and associated activities will be identified. Moreover, an evaluation plan will be developed and established.*


The intervention planning will proceed based on the logic model. Project milestones with according activities will be identified by the HeLP management team for intervention development, implementation, and evaluation. Materials, training, and processes will be purchased or developed in collaboration between the participating sites and the HeLP management team.

Step 6. Intervention development and refinement

**Step** **6.**
*Focuses on intervention development and refinement including performing a series of quality improvement cycles to test materials, training, manuals, and processes. Content will be refined based on findings from these cycles.*


Quality improvement cycles [50] will be undertaken by the HeLP management team to test intervention elements including materials, training, and manuals. We anticipate that quality improvement cycles will be based on the Plan-Do-Study-Act (PDSA) method [50,51].

Under ‘plan’, ideas for improvements related to materials, training, or manuals in HeLP will be detailed, responsibility and task assignments will be established, and expectations will be discussed and agreed upon. Moving on to ‘do’, where the plan is implemented and tested. Under ‘study’, any deviations or defects detected during the do phase will be analyzed and studied. Finally, under ‘act’, the learnings generated will be incorporated into the element, which was tested. Based on findings from these cycles, materials and processes will continuously be refined.

### 2.5. Phase Three

Step 7. Implementation and evaluation

**Step** **7.**
*Focuses on refining, implementing, and evaluating the intervention.*


In phase three, the first activity is to refine the implementation and evaluation plan. The intervention will be implemented followed by evaluation activities. Content and suitable methods used for the evaluation strategy will depend on the developed intervention and will therefore not be established before the development process is finalized. We anticipate that qualitative and mixed methods will be used to evaluate the intervention due to the potential complexity and context-bound nature [52]. The evaluation strategy will be developed with the objective to gain contextualized understandings of how the intervention works, and how it changes outcomes in practice. We expect that the evaluation strategy will contain qualitative feasibility studies [53,54]. Some of the following questions will potentially be suitable to guide the evaluation strategy in HeLP. These preliminary questions were inspired by a qualitative version of the RE-AIM (Reach, Effectiveness, Adoption, Implementation, and Maintenance) framework [55,56], a framework based on five dimensions to evaluate public health interventions.

Reach: What factors contribute to participation/non-participation? Does the intervention reach the participants, who needs it most?Effectiveness: Does the intervention have a meaningful effect and benefit for participants? Are there any unanticipated outcomes of the intervention? Does the intervention work with typical pregnant women and health professionals in a real-world setting? Are the results meaningful?Adoption: Are interventions adopted at all organizational levels of the participating sites, by health professionals, and pregnant women? What barriers reduce intervention adoption?Implementation: Is the intervention delivered as intended by selected health professionals? By whom and when was the intervention implemented? What influenced implementation or lack of implementation?Maintenance: Is the intervention institutionalized as part of the everyday culture and norms at the participating sites?

The final outcome measures and methods used to develop an evaluation strategy in the HeLP program will be established by the HeLP management team when the intervention is developed.

Step 8. Development of an ongoing quality improvement strategy

**Step** **8.**
*Focuses on quality improvement. Intervention components will undergo continuously quality improvement in step 8 of the HeLP program.*


An ongoing quality improvement strategy [57] will be developed based on the Plan-Do-Study-Act (PDSA) method [50,51], as described under step 6.

## 3. Ethics

The HeLP program is approved by the Danish Data Protection Agency (2016-051-000001, 2296) and the Regional Ethics Committee. Informed consent will be collected from consultation workshop participants to audio-record table dialogue. The questionnaire will be created in the data-protected system, Research Electronic Data Capture (RedCap). Data will be cleansed, quality ensured, and anonymized by ID numbers.

## 4. Discussion

The HeLP program is expected to result in new knowledge of pregnant women’s health literacy needs, as well as the development, implementation, and evaluation of a health literacy intervention for antenatal care, which has the potential to accommodate and respond to different levels of health literacy among pregnant women. We expect that the application of the Ophelia process will guide the co-design process successfully, entail large engagement from stakeholders and increase practical outcomes, which will benefit pregnant women, their partners, and their unborn child [30,31,33].

### 4.1. Strengths and Limitations

The HeLP program involves two sites: both a tertiary referral hospital and a secondary hospital. The tertiary hospital is located in Aarhus—the second largest city in Denmark (approximately 340,000 residents), while the secondary hospital is located in Viborg—a smaller Danish city (approximately 41,000 residents). The involvement of different participating sites in the HeLP program is a strength as it increases the external validity of findings. Due to the co-design process based on local knowledge, findings may not directly be reproduced in other contexts, and local adaptions may always be considered before scaling up [58].

Today, antenatal care is complex and based on a high level of patient self-management, e.g., in relation to technology use and navigation [59]. The complexity increases the risk of inequality in health and the gap between highly and less resourceful patients [60]. In the HeLP program, we seek to involve a broad variety of local stakeholders including pregnant women in vulnerable life situations. It is a strength of the HeLP program that workshop participants will be recruited from different settings and represent a heterogeneous group. Moreover, the Ophelia process has been tested in various settings and was found suitable for the successful involvement of local stakeholders including participants in vulnerable life situations [32,33]. However, we also foresee some challenges and weaknesses in the HeLP program. Engaging in a scientific project and filling out a comprehensive questionnaire requires basic health literacy skills [61]. Hence, an important group potentially eludes participation and filling out HeLP-Q. We are aware of this challenge and plan to have research assistants present at participating sites to help and support pregnant women in filling out the questionnaire.

The co-creative methodology used in the HeLP program has the potential to generate empowering processes (enable participants to gain control, develop skills, and test their knowledge) and empowering outcomes (a feeling of increased control, greater understanding, and active involvement) among participants [62]. On the other hand, some literature suggests that highly resourceful participants may dominate the co-design processes, due to their superior capital [62]. The HeLP management team needs to acknowledge and consider this challenge. For the co-design process in HeLP to succeed and produce useable and relevant practical findings, consideration of participant’s motivation and professional-patient and inter-participant relationships and roles is important [62]. The HeLP management team, who is responsible for the workshops, collaboration with participants, and communication, must demonstrate an open attitude towards participants. If participants feel mistrusted by the researchers, the risk of negative attitudes towards the co-design process increases [62,63]. The researchers hold the main responsibility to set the frame, plan, structure, and organize meetings, workshops, and other collaborative activities with participants. However, they must be aware of their role and support and endorse participant empowerment [64].

The Ophelia process allows for some adaption during the process. For example, consultation workshops are scheduled to last approximately four hours each and include 25 participants. These decisions were made based on experiences from previous Ophelia processes [32] and a professional judgment. We do not know whether this timeframe and participant number are realistic and suitable in practice. If the HeLP management group experiences any challenges related to this during the first workshop, adjustments will be made before the second workshop. We will try to accommodate any challenges related to this and provide auxiliary assistance during the workshops.

Another limitation of the HeLP program is that health literacy interventions developed based on the Ophelia process need further investigation and evaluation to establish effectiveness. However, preliminary findings are promising [33].

### 4.2. Implementation of Findings

The development of health literacy interventions for antenatal care is important due to increasing inequality in access to antenatal care services among pregnant women [1,2,4]. A health literacy focus is needed to support and meet pregnant women’s different skillset to access, understand, appraise, and apply health information and make decisions related to the health of themselves and their baby. These skills have only become more important due to the increasing complexity, and need for engagement, decision making, and use of technology in health care services [65].

In 2017, Beauchamp et al. published a paper on the systematic development and implementation of interventions to optimize health literacy and access [30]. The study showed improvements in health literacy scores, and the Ophelia process was successfully applied resulting in the development of health literacy interventions in accordance with local wisdom and organizational priorities [30]. Evidence of operationalization of the eight Ophelia principles was present at all intervention sites. Hence, (1) the operationalization was outcome focused, (2) the sites were equity driven, (3) and (5) the wisdom of local stakeholders ensured co-design, (4) and (7) intervention ideas drew on local wisdom sites focused on local health literacy needs, and (8) ideas were generated and applied across all levels of organizations. However, (6) the sustainability of the interventions needs confirmation in a long turn follow-up, as well as the improvement processes to ensure interventions remain relevant and effective [30].

In addition, Jessup et al. used the Ophelia process in hospitalized populations [66], while Anwar et al. applied the process in fishing villages in Egypt [67]. Moreover, Cheng et al. used the process with an eHealth Literacy focus [68]. Similar to these studies is that they contribute evidence supporting that the Ophelia process produces user-friendly vignettes and provides a locally driven and contextual co-design process [66,67,68]. However, evidence of the implementation and evaluation process in Ophelia needs further exploration as well as the sustainability of produced health literacy interventions and the ongoing process to ensure relevance and effectiveness of the interventions developed using the Ophelia process.

The HeLP program is expected to contribute new knowledge about how the Ophelia process can be used to develop health literacy interventions, which generate improvements in antenatal care and address inequity in access to services [30]. In addition, we anticipate that the HeLP program will contribute further knowledge on the effectiveness of the Ophelia process, and the operationalization of the eight Ophelia principles in antenatal intervention sites.

## 5. Conclusions

The HeLP program is expected to contribute new knowledge of pregnant women’s health literacy needs in antenatal care and development, implementation, and evaluation of health literacy interventions. The Ophelia process will be used to co-design health literacy interventions based on local knowledge about health literacy strengths and challenges among pregnant women and improve the ability to respond to these strengths and challenges in antenatal care.

## Figures and Tables

**Figure 1 ijerph-19-04449-f001:**
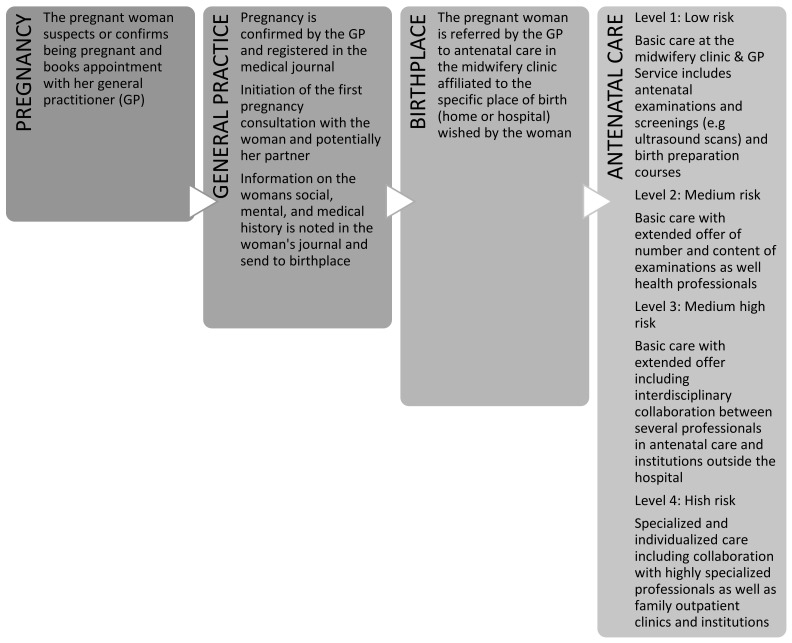
Organization of Danish antenatal care.

**Figure 2 ijerph-19-04449-f002:**
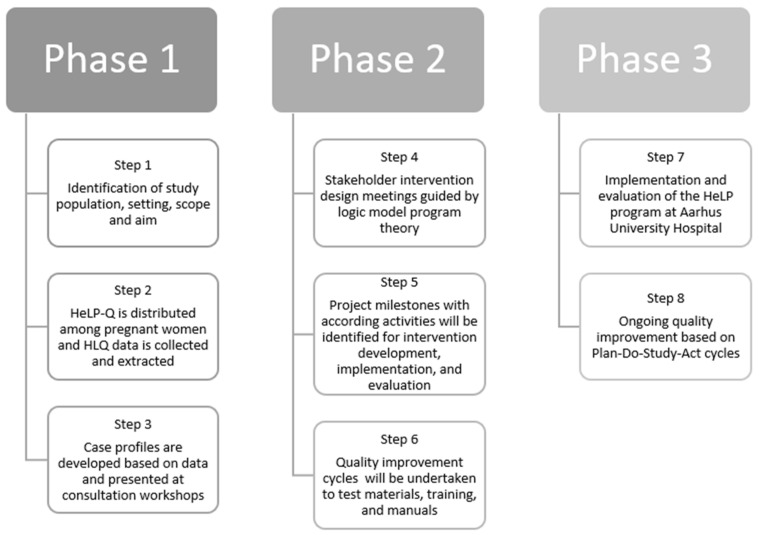
The Ophelia process—three phases and according steps in the HeLP program.

**Figure 3 ijerph-19-04449-f003:**
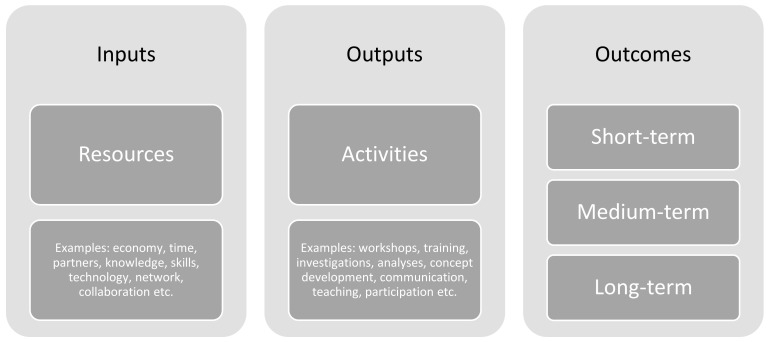
Outline of logic model for interventions in the HeLP program.

**Table 1 ijerph-19-04449-t001:** Preliminary consultation plan, HeLP.

Plan Elements	Content and Arrangements
Time Frame	Approximately Four Hours
Staff responsible	Associate Professor, PhD, and Midwife R.D.M, PhD Fellow, M.M. and Midwife, M.F.D.
Format and participants	Approximately 25 participants for each workshop placed in 5 different discussion groups. Five participants in each group have shown to be ideal in health research focus groups [48] Two workshops held with pregnant women, partners, family members, healthcare providers and health professionals including obstetricians, general practitioners, midwives, nurses, social workers, psychologists, NGO-employees, and other stakeholders First workshop: results from HeLP-Q and case profiles will be presented, and participants will be instructed to make table mindmaps with intervention ideas Second workshop: participants will be instructed to discuss and prioritize an intervention package based on the matched intervention ideas and objectives
Recruitment approach	Pregnant women and partners will be invited to participate in workshops from the two participating sites We plan to recruit from different settings including basic midwifery consultation and other related services which provides care for pregnant women with challenges related to physical, mental, or social health, socio-economic factors, etc. The participating group of pregnant women and partners should preferably include a heterogeneous group Health professionals (a broad variety of professionals working in or with antenatal care) will be invited for workshop participation from different organizational levels at the two participating sites We plan to recruit health professionals, who work in different settings and organizational levels of antenatal care
How to capture ideas and insights	Four research assistants and three students will be present at workshops to observe and take notes A table manager, who are responsible for writing down during workshops will be assigned for each table Table managers will be instructed to fill out table mindmaps summing up all ideas and thoughts Informed consent will be sent by email and signed by participants before workshop days, and dialogues at each table will be audio recorded

## Data Availability

Not applicable.

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
