# Peer review of "The Health Literacy in Pregnancy (HeLP) Program Study Protocol: Development of an Antenatal Care Intervention Using the Ophelia Process"

_ijerph, 2022, doi:10.3390/ijerph19084449_

Round 1

Reviewer 1 Report

The manuscript is an extremely abundant, professional, and promising summary of a methodology for examining the extent of health literacy during pregnancy. The paper is well documented, easy to understand, and the range of references is sufficient. However, the reviewer feels like the restaurant guest who paid for the entire menu but only the soup was served. The protocol should have been tested on at least a smaller sample. I’m not sure that the IJERPH readership is interested in such somewhat one-sided communications. I think that the results of such studies must also be seen in order to realistically assess the effectiveness and applicability of the methodology. Therefore, despite its undoubted values.

Author Response

Dear reviewers,

Thank you very much for your time and useful comments/suggestions for this paper. We have accommodated your suggestions and think the revision have improved the paper. Below, please find your comments listed with our replies added in green. Changes in the manuscript has been made with ‘track changes’.

Kind regards,

Maiken Meldgaard, corresponding author

Reviewer 1:

Comments and Suggestions for Authors

The manuscript is an extremely abundant, professional, and promising summary of a methodology for examining the extent of health literacy during pregnancy. The paper is well documented, easy to understand, and the range of references is sufficient. However, the reviewer feels like the restaurant guest who paid for the entire menu but only the soup was served. The protocol should have been tested on at least a smaller sample. I’m not sure that the IJERPH readership is interested in such somewhat one-sided communications. I think that the results of such studies must also be seen in order to realistically assess the effectiveness and applicability of the methodology. Therefore, despite its undoubted values.

Author’s reply:

Thank you very much for your kind words. We are very happy to see that you find the introduction/background sufficient, the research design appropriate and the methods described adequately. We agree that no test results are presented as this is ‘study protocol’. We agree that pilot studies are also relevant to conduct before original studies. However, we argue that the manuscript still has a place and will be valuable to the readers of IJERPH. The protocol provides a detailed description of the development phase of a health intervention using the Ophelia approach and co-creation. It gives a thorough description of how patient and public involvement can be done and may help others to facilitate similar processes. We find that sharing this development protocol and providing transparency of how we are going to use the methodology into a specific project will inspire and help other researchers working with complex interventions. In addition, despite comprehensive literature about how to develop complex interventions, an intervention development phase protocol using the Ophelia approach has, to the best of our knowledge, not previously been published. The effectiveness and applicability of the methodology have shown promising in other studies using the Ophelia approach, which we refer to in reference (1-4) and we will test and evaluate the HeLP program thoroughly once the program has been implemented.

  1. Aaby A, Simonsen CB, Ryom K, Maindal HT. Improving Organizational Health Literacy Responsiveness in Cardiac Rehabilitation Using a Co-Design Methodology: Results from The Heart Skills Study. International journal of environmental research and public health. 2020;17(3).
  2. Batterham RW, Buchbinder R, Beauchamp A, Dodson S, Elsworth GR, Osborne RH. The OPtimising HEalth LIterAcy (Ophelia) process: study protocol for using health literacy profiling and community engagement to create and implement health reform. BMC public health. 2014;14:694.
  3. Beauchamp A, Batterham RW, Dodson S, Astbury B, Elsworth GR, McPhee C, et al. Systematic development and implementation of interventions to OPtimise Health Literacy and Access (Ophelia). BMC public health. 2017;17(1):230.
  4. World Health Organization. Regional Office for E, Bakker MM, Putrik P, Aaby A, Debussche X, Morrissey J, et al. Acting together – WHO National Health Literacy Demonstration Projects (NHLDPs) address health literacy needs in the European Region. Public health panorama. 2019;5(2-3).

Reviewer 2 Report

Title: The Health Literacy in Pregnancy (HeLP) program study protocol: Development of an antenatal care intervention using the Ophelia approach

The paper provides a good narrative on the development protocol for the HeLP program, pregnant women’s health literacy and health literacy interventions in collaboration between pregnant women, healthcare providers, professionals, and other stakeholders using the proposed program. The affinity diagrams are well defined.

The authors make a good point that Health literacy is associated with socio-economic factors and seem to follow a social gradient. However, they could provide a clear explanation as to what is a social gradient. The paper is well written and contains useful information. I have a few minor points the authors may wish to consider.

  1. The authors could describe the key terms in detail.
  2. Authors make assumptions on the application of the program in two hospitals, could explain more on the selection criteria of these two hospitals? Aarhus University Hospital and a secondary hospital, Regional Hospital in Viborg.
  3. “The Ophelia (OPtimising HEalth LIteracy and Access) approach is used to guide co-creation of interventions to improve health literacy and equity in healthcare services” what is meant by co-creation?
  4. The term is confusing: “The Ophelia (OPtimising HEalth LIteracy and Access). Can be OELA
  5. Data collection methods needs explanation: Phase one: Identification of local strengths, needs, and issues includes Step 1) project set-up Step 2) data collection and/or extraction
  6. What is the criterion for creation of the 5 discussion groups in Table 1?

Author Response

Dear reviewers,

Thank you very much for your time and useful comments/suggestions for this paper. We have accommodated your suggestions and think the revision have improved the paper. Below, please find your comments listed with our replies added. Changes in the manuscript has been made with ‘track changes’.

Kind regards,

Maiken Meldgaard, corresponding author

Reviewer 2:

Comments and Suggestions for Authors

Title: The Health Literacy in Pregnancy (HeLP) program study protocol: Development of an antenatal care intervention using the Ophelia approach

The paper provides a good narrative on the development protocol for the HeLP program, pregnant women’s health literacy and health literacy interventions in collaboration between pregnant women, healthcare providers, professionals, and other stakeholders using the proposed program. The affinity diagrams are well defined.

The authors make a good point that Health literacy is associated with socio-economic factors and seem to follow a social gradient. However, they could provide a clear explanation as to what is a social gradient. The paper is well written and contains useful information. I have a few minor points the authors may wish to consider.

Author’s reply: Thank you very much for your comments. We are pleased to read that you find that the paper provides a good narrative, and that the affinity diagram are well defined.

  1. The authors could describe the key terms in detail.

Author’s reply: We agree that ‘social gradient’ could be described in more detail to the readership. Thank you for pointing this out. We changed the manuscript line 44-47 and added details: and seem to follow a social gradient; a phenomenon whereby people who are less advantaged in terms of socioeconomic position have poorer health compared to those who are more advantaged (5, 6)

  1. Authors make assumptions on the application of the program in two hospitals, could explain more on the selection criteria of these two hospitals? Aarhus University Hospital and a secondary hospital, Regional Hospital in Viborg.

Author’s reply: Thank you for your insights. We agree that the selection criteria could be explained in more detail. We added details in line 81-84: The two sites handle the midwifery consultations during antenatal care. The two intervention sites differ in size, location, and organization. The inclusion of both sites is an attempt to increase the representativeness of the study population.

  1. “The Ophelia (OPtimising HEalth LIteracy and Access) approach is used to guide co-creation of interventions to improve health literacy and equity in healthcare services” what is meant by co-creation?

Author’s reply: Thank you for pointing this out! We have added details in the manuscript and describes co-creation more thoroughly to the readership. We changed line 58-62: The Ophelia (OPtimising HEalth LIteracy and Access) approach is used to guide co-creation – a method to involve and engage relevant stakeholders into the process - of interventions to improve health literacy and equity in healthcare services (7).

  1. The term is confusing: “The Ophelia (OPtimising HEalth LIteracy and Access). Can be OELA

Author’s reply: We are sorry that you find the term confusing. However, the name and abbreviation were chosen by the research team that developed the approach originally, and we are therefore not able to change the name.

  1. Data collection methods needs explanation: Phase one: Identification of local strengths, needs, and issues includes Step 1) project set-up Step 2) data collection and/or extraction

Author’s reply: Thank you for the comment. Phase one including data collection methods is adapted and outlined specifically for the HeLP program in figure 2 and line 143-178.

  1. What is the criterion for creation of the 5 discussion groups in Table 1?

Author’s reply: Thank you for this input. We changed the line in table 1 and added more detail. See the following line from table 1.

Approximately 25 participants for each workshop placed in 5 different discussion groups. Five participants in each group has shown to be ideal in health research focus groups (8).

  1. Bo A, Friis K, Osborne RH, Maindal HT. National indicators of health literacy: ability to understand health information and to engage actively with healthcare providers - a population-based survey among Danish adults. BMC public health. 2014;14:1095.
  2. Sørensen K, Pelikan JM, Röthlin F, Ganahl K, Slonska Z, Doyle G, et al. Health literacy in Europe: comparative results of the European health literacy survey (HLS-EU). European journal of public health. 2015;25(6):1053-8.
  3. Hardyman W, Daunt KL, Kitchener M. Value Co-Creation through Patient Engagement in Health Care: A micro-level approach and research agenda. Public Management Review. 2015;17(1):90-107.
  4. Tausch AP, Menold N. Methodological Aspects of Focus Groups in Health Research: Results of Qualitative Interviews With Focus Group Moderators. Global qualitative nursing research. 2016;3:2333393616630466-.

Round 2

Reviewer 1 Report

I understand and appreciate the authors ’argumentation, but since the manuscript has changed only minimally, I maintain my opinion. I see at least one pilot study worth testing the carefully crafted methodology. The investigation protocol is exciting and excellent, but the communication in this form is one-sided. 

I still don't recommend publishing the paper as it stands. However, I do not object to the manuscript being published for editorial reasons.

Author Response

Author’s reply:

Thank you for your second review of the paper. We agree that a pilot study would have been of value to the HeLP project and our publication. This have, however, not been feasible under the circumstances. However, we feel assured by the many other projects that have used the Ophelia methodology with success in a range of different settings and target groups. This have been further highlighted in our manuscript by updating the included literature for development papers using the Ophelia process. We have added a section under ‘discussion’ from line 335-352, which includes papers that have tested the methodology in other settings and provide results from the Ophelia process. These results are used in the discussion section to support the findings the HeLP program is expected to provide. We hope that you will find that the updated section accommodates the problem of one-sided communication.

Reviewer 2 Report

Improve the abstract and include the result details.

Author Response

Author’s reply:

Thank you for providing a second review of the paper. We have added a section under ‘discussion’ from line 335-352, which include papers that have tested the methodology in other settings and provides outcomes from the Ophelia process. These results are used in the discussion section to support the findings the HeLP program is expected to provide. We hope you find that the added section supports the expected results of the HeLP program.